# High-Intensity Interval Training in Female Adolescents with Moderate or Severe Obesity

**DOI:** 10.3390/children10091495

**Published:** 2023-09-01

**Authors:** Ghazi Racil, Luca Russo, Gian Mario Migliaccio, Paola Signorelli, Alin Larion, Johnny Padulo, Mohamed Chedly Jlid

**Affiliations:** 1Research Unit (UR 17JS01) “Sport Performance, Health & Society” Higher Institute of Sport and Physical Education of Ksar Said, Tunis 1000, Tunisia; ghazi_racil@yahoo.fr (G.R.); chedly3@yahoo.fr (M.C.J.); 2Department of Biological Sciences Applied for Physical Activities and Sport, Higher Institute of Sport and Physical Education of Ksar Said, University of La Manouba, Manouba 2010, Tunisia; 3Department of Human Sciences, Università Telematica degli Studi IUL, 50122 Florence, Italy; l.russo@iuline.it; 4Department of Human Sciences and Promotion of the Quality of Life, San Raffaele Rome Open University, 00166 Rome, Italy; gianmario.migliaccio@uniroma5.it; 5Institute for Molecular and Translational Cardiology (IMTC), San Donato Milanese, 20097 Milan, Italy; paola.signorelli@unimi.it; 6Aldo Ravelli Center for Neurotechnology and Experimental Brain Therapeutics, Department of Health Sciences, University of Milan, Via Antonio di Rudinì 8, 20142 Milan, Italy; 7Faculty of Physical Education and Sport, Ovidius University of Constanta, 900029 Constanta, Romania; alinlarion@yahoo.com; 8Department of Biomedical Sciences for Health, Università degli Studi di Milano, 20133 Milan, Italy

**Keywords:** physical activity, adolescent, HIIT, insulin resistance index, growth hormone

## Abstract

This study aimed to investigate the impact of moderate- or high-intensity interval training (MIIT or HIIT) on anthropometric and biological measurements in four groups of females with obesity. Fifty-seven participants were divided into a moderate obesity group (MOG, *n* = 29) and a severe obesity group (SOG, *n* = 28). Two sub-groups were established to practice HIIT and MIIT programs (SOG_HI_, *n* = 14; SOG_MI_, *n* = 14; MOG_HI_, *n* = 14; MOG_MI_, *n* = 15). During the training sessions, each group performed two sets of 4 × 1 min intervals on a cycle ergometer. The intervals were conducted at 65% and 85% of the heart rate reserve (HRR) for MIIT and HIIT, respectively. Between each repetition, there was an active recovery phase at 50% HRR, and, between sets, there was a 4 min period of free pedaling. All groups significantly improved their anthropometric data, while only MOG_HI_ and SOG_HI_ significantly improved their lean body mass (LBM) and blood lactate (BL), with *p* ˂ 0.05; the higher percentage of change in blood insulin levels (−25.49 and −25.34) and the homeostasis model assessment of the insulin resistance index (−31.42 and −28.88) were noted. Only MOG_HI_ showed improvements in growth hormone (GH) and blood glucose (*p* < 0.05), which were negatively correlated with body fat percentage (r = −0.76 and r = −0.72) and waist circumference (r = −0.77 and r = −0.82), respectively. We may conclude that HIIT was an effective method of managing anthropometric and biological parameters, as confirmed by the pronounced body fat reduction in the moderate obesity group.

## 1. Introduction

Obesity has become an epidemic disease that is associated with a variety of comorbidities. The most implicated factors involve the simultaneous interactions among genetics, biology, the environment, and behavior [1]. Furthermore, it is known that children’s low levels of physical activity and high amounts of time spent engaging in sedentary behavior lead to various cardiovascular complications and these tend to be more prevalent in overweight and obese children [2] because of the increased level of body fat [3]. While this is the situation, physical activity is becoming an important determinant in the treatment of childhood obesity and early metabolic risk factors [4]. In the past, continuous exercise was often offered as an efficient method for rehabilitation; subsequently, moderate-intensity interval training showed its greatest physiological benefits [5].

Thus, a broad range of strategies are recommended to reduce the increasing prevalence of obesity, including running exercises at various intensities. High-intensity interval training (HIIT), which depends on repeated high-intensity exercises alternated with short recovery periods, is considered a time-efficient exercise strategy to promote physical fitness and health compared to other training methods [6,7]. HIIT was found to improve the adaptations in several skeletal muscles, resulting in an enhanced fat oxidative capacity and improved glucose transport, which led to improved glycemic control, whether after long- [8] or short-term training periods [9]. It is important to consider that such enhancements could further improve insulin sensitivity [8,9] and reduce some cardiometabolic risk factors in obese youth [10]. Recent results [11] suggested that HIIT is useful both in the short term and long term for improving blood and anthropometric parameters both in moderately and in severely obese male adolescents. Moreover, the percentage of change was higher in severely obese male adolescents who trained with HIIT [11]. Although these results can be considered to be very interesting, there are a lack of data about female adolescents, particularly for the hormonal profile such as the growth hormone (GH). The GH is a peptide hormone that is secreted by somatotrophs in the anterior pituitary gland and has a variety of tissue-specific effects, including anabolic effects on muscle and bone and catabolic action on white adipose tissue [12]. Generally, the GH gradually increases during childhood, achieving a peak during puberty, and then declines during adult life [13,14]. It is important to consider that the GH acts on glucose, proteins, and lipids metabolism [12,15]. Even after training, the GH secretion in an obese person sometimes remained reduced [14], but some other studies showed that endocrine adaptations are associated with reductions in weight in obese children [16]. This fact seems based on the process of storage and the release of triglycerides, which are controlled in part by the GH, from adipose tissue [12]. Moreover, it was demonstrated that exercise training induces an increase in GH levels in individuals with abdominal obesity, which showed reduced GH levels prior to exercising [17]. Thus, investigating what relation exists between anthropometric and GH levels is worthwhile. According to this framework, decreasing the amount of body fat appears to be of importance. To our knowledge, no specific data are available about the effects of interval training programs in young obese females from different obesity classes. In fact, whether the growth hormone concentration is similarly triggered by intense exercise in obese people with different amounts of body fat is yet to be determined.

Therefore, the purpose of the current study was to compare the effects of moderate- or high- intensity interval training (MIIT or HIIT) on anthropometric and biological data in female adolescents from different obesity classes. According to previous research [11], we hypothesized that HIIT would yield notably superior results when enhancing the anthropometric and biological measurements among adolescents, particularly those with moderate obesity, compared to MIIT.

## 2. Materials and Methods

### 2.1. Participants

Fifty-seven obese girls (age 16.4 ± 0.8 years) were recruited for this study. These participants were middle school students invited by their physical education teachers. They were classified according to their body mass index (BMI) using the algorithm provided by the Centers for Disease Control and Prevention. BMI assessments were conducted by a trained school pediatrician. Each participant was required to have a BMI ≥ 97th percentile according to French standards [18], as well as a percentage of body fat (%BF) ≥ 30%. To form the intervention groups, subjects were randomized and stratified based on their age and BMI. Participants were categorized as moderately obese (MOG, *n* = 29) if their BMI ranged between 30.0 and 34.9 kg·m^−2^, and as severely obese (SOG, *n* = 28) if their BMI ranged between 35.0 and 39.9 kg·m^−2^. Within each BMI class, two sub-groups were established to participate in either a HIIT or a MIIT program (SOG_HI_, *n* = 14; SOG_MI_, *n* = 14, and MOG_HI_, *n* = 14; MOG_MI_, *n* = 15).

The means and standard deviations (SD) for BMI, %BF, and body mass (BM) were as follows: 34.1 ± 1.6 kg·m^−2^, 38.6 ± 3.9%, and 95.1 ± 4.5 kg, respectively.

### 2.2. Anthropometrical Measurements

During the first session, height was assessed in centimeters without shoes, with the heels together and with the participant’s back parallel to the stadiometer (Model height rod, Seca®, Hamburg, Germany). Body mass (BM) was measured to the nearest 0.1 kg, and the percentage of body fat (%BF) and lean body mass (LBM) were assessed via bioelectrical impedance analysis (TBF-300, Tanita®, Tokyo, Japan). The body mass index (BMI) was subsequently calculated using the formula: BMI = Mass (kg)/Height (m)^−2^. To estimate central body fat [5], waist circumference (WC) was measured in centimeters using a non-deformable tape ruler, positioned between the lower rib margin and the iliac crest. The pubertal stage was self-assessed by participants using Tanner’s puberty rating scale, a method with recognized validity and reliability [19].

During the same session, the resting heart rate (HR_rest_ in bpm) was measured after a 10-minute sitting period using a heart rate monitor (S-610, Polar®, Kempele, Finland). The lowest heart rate recorded during this period was considered to be the HR_rest_. The maximal heart rate (HR_max_) was estimated for each participant using the following equation [20]: HR_max_ = 206.9 − (0.67 × age). Both HR_rest_ and HR_max_ were utilized to prescribe training session intensities, based on the heart rate reserve.

(HR_reserve_ = HR_max_ − HR_rest_), by using the following equation: = [(HR_max_ − HR_rest_) × % Intensity] + HR_rest_ [21].

### 2.3. Blood Measurements

In the second session, blood samples were collected after a 20-min resting period in a lying position. The samples were centrifuged at 4 °C, and the plasma was stored at −80 °C until analysis. The concentration of growth hormone (GH) was determined using an enzyme-linked immunosorbent assay (ELISA) kit (Diagnostic Biochem Canada Inc., London, Ontario, Canada). The intra-assay and inter-assay coefficients of variation (CV) were 4.8% and 11.4%, respectively. Lactate concentration was measured using an automated ultraviolet enzymatic method (Hitachi 717, Roche Diagnostics®, Laval, QC, Canada) with a CV of <8%. Glucose concentration was assessed using an automated device (AU2700, Olympus®, Rungis, France), and the inter-assay CV was 1.8%. Insulin concentration was assayed using an IRMA insulin kit (Immunotech®, Marseille, France). The intra- and inter-assay CV were 3.2% and 4.8%, respectively.

Insulin resistance was estimated using the homeostasis model assessment of insulin resistance (HOMA-IR) index, which has been validated for children and adolescents [22].

### 2.4. Group Seletion Test

During the third session, participants underwent a submaximal exercise test to assess their capability to safely perform HIIT. This familiarization session was conducted for each subject in the morning, following a standardized breakfast (10 kcal·kg^−1^, consisting of approximately 55% carbohydrates, 33% lipids, and 12% proteins). The breakfast was formulated by a nutritionist to minimize circadian variations [23,24].

### 2.5. Inclusion and Exclusion Criteria

Participants were required to be free of any lower-body injuries and have no medical, cardiovascular, or orthopedic dysfunctions. None of the participants reported any health issues, nor were they undergoing any drug or therapeutic treatments for obesity. Additionally, their physical activity for the preceding three months had been limited to two hours of school physical education.

To be included, participants had to successfully perform a submaximal ramp incremental exercise test on a cycle ergometer (U60 Upright Bike, Vision Fitness®, Taichung, Taiwan) equipped with an electrical brake system. Following a 5-min rest period, the exercise commenced with a 3-min warm-up at 20 W. The load was then increased by 20 W·min^−1^ and maintained at 50–60 rpm until the target percentage of HR_reserve_ was reached (i.e., the highest intensity prescribed during the initial four weeks of HIIT). It should be noted that the initial sample included 62 participants—31 in each of the SOG and MOG groups. Three from the SOG and two from the MOG were subsequently excluded after verifying their ability to perform HIIT safely.

Both parents and adolescents were informed about the aim and procedures of the study, as well as its potential risks and benefits. Additionally, this study received approval from our Institution’s Research Ethics Committee (No. 44, dated February 18, 2022). Verbal consent was obtained from the adolescents, and written permission was secured from their parents, in accordance with international ethical standards and, more specifically, the Declaration of Helsinki.

### 2.6. Training Programs

Training programs commenced immediately after the completion of all initially planned tests. All groups engaged in 12-week HIIT (SOG_HI_ and MOG_HI_) or MIIT (SOG_MI_ and MOG_MI_) programs, meeting three times per week (Monday, Wednesday, and Friday). For the initial period (from the 1st to the 4th week), participants in both the HIIT and MIIT groups performed 2 sets of 4 × 1 min bouts at intensities of 85% and 65% HR_reserve_, respectively. Two minutes of active recovery between repetitions at 50% HR_reserve_ were allotted to all groups. Between sets, participants engaged in free pedaling without load at a self-selected and constant pace for 4 min. After each 4-week period, the target heart rate was increased by 5% for both HIIT and MIIT groups (Table 1). Each group commenced sessions with a 5 min warm-up at their own pace, not exceeding 50% HR_reserve_. At the conclusion of each training session, subjects participated in approximately 5 min of static stretching for cooldown.

Two members of our research team supervised all testing procedures and training sessions. All training was conducted on the same cycle ergometer (U60 Upright Bike, Vision Fitness®, Taichung, Taiwan), and the heart rate was monitored using the same device (S-610, Polar®, Kempele, Finland). The room temperature was maintained between 22 and 24 °C throughout all sessions. Participants were not required to alter their sleeping, eating, or drinking habits for the study’s duration. All measurements and tests were conducted before and after the 12-week intervention period, under identical testing conditions.

### 2.7. Statistical Analysis

The descriptive data are presented as the mean and standard deviation (SD). Shapiro–Wilk and Levene’s tests were utilized to verify the normal distribution of data and the homogeneity of variances, respectively. An analysis of variance (ANOVA) was employed to compare baseline characteristics among the four groups [11].

Since the data were normally distributed, a two-way analysis of variance (ANOVA) with repeated measures was used to compare values within each group (SOG and MOG) for each training method (HIIT and MIIT). For within-group changes, a paired-sample *t*-test was employed to identify differences between pre-intervention and post-intervention values. Interclass correlation coefficients (ICC) were calculated to evaluate the reliability of all body composition measurements. When a significant interaction effect was detected, Tukey’s honest significant difference test was applied. 

An analysis of covariance (ANCOVA) was conducted to assess the influence of Tanner stages on significant differences both within and between groups. Effect sizes (ESs) for the time × group interaction effects were calculated using partial eta-squared (η^2^). These ES were subsequently converted from the ANOVA output to Cohen’s d. 

The ES served to estimate the magnitude of the difference (i.e., trivial: ES < 0.2; small: 0.2 < ES < 0.5; moderate: 0.5 < ES < 0.8; and large: ES > 0.8). The level of significance was set at *p* < 0.05. The correlation between GH and other parameters was examined using Pearson’s test (r). Statistical analyses were performed using SPSS (IBM SPSS Statistics for Windows, Version 24.0, Armonk, NY, USA: IBM Corp).

## 3. Results

Anthropometric and biological data, gathered before and after the intervention programs, are displayed in Table 2 and Table 3. For all body composition measurements, the reliability was excellent, with an ICC exceeding 95%. Comparisons between the two sub-groups in each BMI class prior to the intervention revealed that they were well matched in terms of age and anthropometric parameters.

In the post-intervention analysis, significant reductions were noted across all training groups in BM, BMI, %BF, and WC (*p* < 0.05). These reductions were especially pronounced in MOG_HI_, registering at (*p* < 0.01). The highest percentages of change (Δ) were observed in MOG_HI_ (−2.07, −2.83, −7.52, and −4.92, respectively, as detailed in Table 4). Only the groups training at high intensity exhibited significant increases in LBM (*p* < 0.05), with MOG_HI_ showing the most substantial percentage change (−2.36%).

Additionally, MOG_HI_ was the only group to exhibit a significant increase in GH (*p* < 0.05), as shown in Table 4. Significant decreases in BL values were noted in both SOG_HI_ and MOG_HI_ (*p* < 0.05; ES = 0.61 and 0.56, respectively). A notable reduction in blood glucose occurred exclusively in MOG_HI_ (*p* < 0.05, ES = 0.68), differentiating it significantly from other groups. Regarding blood insulin and HOMA-IR, all groups experienced significant reductions from pre- to post-intervention, with the most pronounced changes observed in MOGHI and SOGHI (*p* < 0.01); ES values were 0.75 and 0.64 for insulin levels, and 0.69 and 0.66 for HOMA-IR levels in MOG_HI_ and SOG_HI_, respectively. When all parameters were adjusted for Tanner stages, the significant differences remained.

Post-intervention, MOG_HI_ demonstrated a significant association between GH and %BF (*p* < 0.001, r = −0.76), as well as between GH and WC (*p* < 0.001, r = −0.77). Similarly, strong associations were found for blood glucose with %BF (*p* < 0.001, r = −0.72) and with WC (*p* < 0.001, r = −0.82).

## 4. Discussion

The primary aim of this study was to compare the effects of HIIT and MIIT on anthropometric and biological data in obese adolescent girls from different obesity classes. The present results demonstrate that both HIIT and MIIT effectively improved body composition in both moderate and severe obesity groups. However, MOG_HI_ showed greater improvements in biological parameters and exhibited the highest percentage of change (Δ) in all tested variables.

In the post-intervention phase, all groups significantly reduced BM, %BF, WC, and LBM. These results align with previous studies in which HIIT significantly reduced %BF and WC [7,11]. Between-group comparisons revealed that MOG_HI_ had the most significant decreases in body composition values, substantiated by a significantly higher Δ in that group. This might relate to its lower initial %BF value, which may have facilitated adaptation to the training program. In contrast, studies with moderate aerobic training have shown minimal or no weight loss in obese subjects [25].

Importantly, a more significant reduction in WC was observed in MOG_HI_, highlighting the effectiveness of HIIT for reducing body fat. This decrease is vital for young obese individuals, as it lessens obesity-related risks [26]. A twelve-week training period was sufficient to elicit improvements, aligning with another study [10].

While all groups exhibited increased GH levels post-intervention, only MOG_HI_ significantly increased its values (Table 3) and showed the highest Δ compared to other groups. The more substantial decrease in body fat in MOG_HI_ might explain this result (−7.52 ± 0.8%). Given that a higher proportion of lean mass may reduce severe obesity risk, the significant increase in LBM in MOG_HI_ and SOG_HI_ is crucial [27]. This helps elevate the basal metabolic rate.

Our results suggest that the initial fat accumulation might have attenuated GH secretion, while the association between body fat and GH levels post-intervention could indicate the efficiency of this training mode. This is particularly valuable for obese individuals, since fat hydrolysis leads to free fatty acid release and their subsequent oxidation [28].

Moreover, the correlation between GH and decreased WC in MOG_HI_ is significant because growth hormone impacts fatty acid metabolism, particularly regarding intra-abdominal fat deposits [24]. This supports the idea that obese individuals might adapt well to HIIT, promoting better body fat oxidation [28].

Previous literature indicates high resting insulin levels in obese children [29]. All groups in our study showed improved insulin levels, but the most significant improvement was in MOG_HI_ (−25.6 ± 6.3%). Our findings align with studies that attribute improvements in insulin sensitivity to reductions in %BF [30].

Additionally, groups training at high intensity showed greater reductions in %BF and HOMA-IR. These changes, we believe, relate to improvements in LBM, which enhances peripheral insulin sensitivity [31]. This is consistent with recent studies showing that HIIT improved LBM and insulin sensitivity in severely obese adolescents [11].

Except for MOG_HI_, no other group significantly reduced blood glucose levels. These reductions could be due to increased GH levels inhibiting glucose excretion into muscle tissue. This is supported by studies that observed no significant changes in blood glucose after HIIT in healthy adolescents [32,33] or obese girls [11]. In contrast, a session of high-intensity exercise, including four 30 s maximal sprints, led to reduced glucose levels [34].

In severe and moderately obese boys, a recent study found significant reductions, which increased with longer training periods [11]. This is explained by the fact that individuals with severe obesity have lower absorption rates of oxygen and glucose in skeletal muscle, both at rest and during exercise [35].

Excess body fat is considered to be one of the main factors preventing obese participants from lowering their blood lactate levels and adapting to recover rapidly after exercise [36]. Interestingly, the current study showed that both MOG_HI_ and SOG_HI_ significantly decreased their blood lactate levels (−6.5 ± 2.4% and −6.3 ± 2.6%, respectively). Generally, during HIIT, blood lactate levels increase and remain elevated for some time after the session’s conclusion. We suggest that the magnitude of the increase in blood lactate levels in the two groups may depend not only on the intensity but also on the training load of the session, as shown in Table 1. Both training modes aim to utilize fat reserves by increasing mitochondrial density and the activity of enzymes involved in fat oxidation [37].

Lastly, we emphasize that both high and moderate exercise intensities improved anthropometric parameters. However, the most significant decrease in %BF was observed with HIIT, as shown in Table 4, in the moderately obese group. This suggests that reducing body weight is essential for young people with obesity, as it may enhance their emotional well-being post-exercise. These feelings that arise after maximal exercise may encourage greater adherence to future training programs, as previously reported [38]. Our hypothesis, posited at the beginning of this research, is supported by the significant correlations observed in the MOG_HI_ group between blood GH levels and %BF, WC, and blood insulin levels. Hence, our results suggest that following a program of HIIT may lead to long-term improvements in blood GH levels and reductions in blood glucose levels across different obesity classifications. We assume that reaching an intensity of 95% HRR by the end of the intervention may have acclimatized adolescents to sufficiently taxing their glycolytic metabolism.

### Limitations

Several limitations should be noted. Firstly, our population did not include both sexes, which might have offered a different type of comparison. However, this should be viewed in the context that the current results are part of a larger study, the first part of which, containing data on male adolescents, has already been published [11]. Secondly, for ethical and methodological reasons, we were unable to measure GH levels at multiple times throughout the day. Diet monitoring was only applied during testing periods. Daily follow-up during the intervention, with calculation of food intake, could have provided additional insights.

## 5. Conclusions

In conclusion, our findings demonstrate that HIIT significantly impacts various biological and anthropometric parameters in moderately and severely obese female adolescents. Therefore, body fat reduction should be prioritized when facilitating their training practice. Alongside the decrease in body fat, which may impact body image, training sessions become more manageable, leading to increased adherence and motivation. 

However, targeting body fat reduction remains a priority in obese populations to facilitate easier training and promote overall health and well-being.

## Figures and Tables

**Table 1 children-10-01495-t001:** Summary of the training programs for all groups: high-intensity interval training (HIIT) and moderate-intensity interval training (MIIT).

	Weeks		
Groups	1–4	5–8	9–12
HIIT program	Warm-up = 5 min free pedaling	Warm-up = 5 min free pedaling	Warm-up = 5 min free pedaling
	2 × (4 × 1-min/2-min)	2 × (5 × 1-min/2-min)	2 × (5 × 1-min/2-min)
	at 85%/50% HR_reserve_	at 90%/50% HR_reserve_	at 95%/50% HR_reserve_
	R2 = 4 min (free pedaling)	R2 = 4 min (free pedaling)	R2 = 4 min (free pedaling)
	Stretching exercises = 5-min	Stretching exercises = 5-min	Stretching exercises = 5-min
	TL: 540ATU	TL: 700ATU	TL: 725ATU
MIIT program	Warm-up = 5 min free pedaling	Warm-up = 5 min free pedaling	Warm-up = 5 min free pedaling
	2 × (4 × 1-min/2-min)	2 × (5 × 1-min/2-min)	2 × (5 × 1-min/2-min)
	at 65%/50% HR_reserve_	at 70%/50% HR_reserve_	at 75%/50% HR_reserve_
	R2 = 4 min (free pedaling)	R2 = 4 min (free pedaling)	R2 = 4 min (free pedaling)
	Stretching exercises = 5-min	Stretching exercises = 5-min	Stretching exercises = 5-min
	TL: 460ATU	TL: 600ATU	TL: 625ATU

TL (training load). Example of training load calculation for HIIT during the first period: TL = [(85 + 50)/2) × 4 × 2] = 540 ATU.

**Table 2 children-10-01495-t002:** Anthropometrical data (mean ± SD) before and after the training programs.

	MOG (*n* = 29)				SOG (*n* = 28)			
	MOG_MI_ (*n* = 15)		MOG_HI_ (*n* = 14)		SOG_MI_ (*n* = 14)		SOG_HI_ (*n* = 14)	
PS (II-III/IV-V)	7/8		7/7		8/6		7/8	
Age (years)	16.8 ± 0.9		16.2 ± 1.2		15.9 ± 1.1		16.7 ± 0.9	
Height (cm)	168.4 ± 4.1	168.5 ± 4.2	165.6 ± 3.0	165.6 ± 3.4	167.0 ± 3.1	167.0 ± 3.2	168.0 ± 3.0	168.1 ± 3.4
BM (kg)	88.4 ± 4.5	87.3 ± 4.4 *	86.7 ± 4.2	84.9 ± 3.7 *^,£^	101.1 ± 5.1	99.3 ± 4.7 *	103.1 ± 5.4	101.1 ± 4.9 *
BMI (kg·m^−2^)	31.4 ± 1.4	30.9 ± 1.4 *	31.8 ± 1.6	30.9 ± 1.3 *	36.6 ± 1.5	35.7 ± 1.4 *	36.6 ± 2.4	35.8 ± 1.9 *
%BF (%)	36.6 ± 4.1	35.1 ± 4.2 *	35.9 ± 4.3	33.2 ± 3.2 *^,£^	40.2 ± 3.7	39.1 ± 4.1 *	40.6 ± 4.6	38.9 ± 4.1 *
WC (cm)	87.9 ± 6.5	84.7 ± 6.5 *	89.4 ± 7.1	85.0 ± 6.9 ^#^	94.6 ± 6.1	92.1 ± 5.8 *	98.3 ± 6.3	94.8 ± 6.4 *
LBM (kg)	51.8 ± 3.4	52.2 ± 3.2	50.7 ± 4.1	51.9 ± 4.3 *	51.8 ± 3.4	52.2 ± 3.2	49.9 ± 3.5	50.8 ± 3.3 *

Values are mean ± SD; significantly different from pre-intervention: (*): *p* ˂ 0.05; (#): *p* ˂ 0.01. Significantly different from the other groups in post intervention: (^£^): *p* ˂ 0.05. SOG_MI_ and SOG_HI_: the groups with severe obesity trained with moderate- or high-intensity interval training; MOG_MI_ and MOG_HI_: the groups with moderate obesity trained with moderate- or high-intensity interval training; BM: body mass; BMI: body mass index; %BF: percentage of body fat; WC: waist circumference; LBM: lean body mass.

**Table 3 children-10-01495-t003:** Blood variables (mean ± SD) before and after the training programs.

	MOG (*n* = 29)				SOG (*n* = 28)			
	MOG_MI_ (*n* = 15)		MOG_HI_ (*n* = 14)		SOG_MI_ (*n* = 14)		SOG_HI_ (*n* = 14)	
GH (ng_·_mL^−1^)	3.4 ± 0.5	3.5 ± 0.4	3.5 ± 0.4	3.75 ± 0.4 *^,£^	2.7 ± 0.5	2.8 ± 0.5	2.8 ± 0.5	2.9 ± 0.6
BL(mmol_·_L^−1^)	1.2 ± 0.4	1.18 ± 0.4	1.2 ± 0.3	1.11 ± 0.2 *	1.3 ± 0.5	1.27 ± 0.3	1.3 ± 0.4	1.2 ± 0.3 *
Glucose (mmol_·_L^−1^)	4.45 ± 0.15	4.20 ± 0.11	4.40 ± 0.17	4.05 ± 0.16 *^,£^	4.50 ± 0.14	4.35 ± 0.13	4.63 ± 0.11	4.44 ± 0.13
Insulin (μU_·_mL^−1^)	19.9 ± 1.5	16.3 ± 1.6 *	20.4 ± 1.5	15.2 ± 0.2 ^#,£^	21.2 ± 1.6	17.3 ± 1.5 *	21.7 ± 1.8	16.2 ± 1.3 ^#^
HOMA-IR	3.93 ± 0.75	3.04 ± 0.90 *	3.98 ± 0.46	2.73 ± 0.42 ^#,£^	4.24 ± 0.35	3.34 ± 0.40 *	4.50 ± 0.52	3.20 ± 0.63 ^#^

Values are mean ± SD; Significantly different from pre-intervention: (*): *p* ˂ 0.05; (^#^): *p* ˂ 0.01. Significantly different from the other groups in post intervention: ^£^: *p* ˂ 0.05. SOG_MI_ and SOG_HI_: the groups with severe obesity trained with moderate- or high-intensity interval training; MOG_MI_ and MOG_HI_: the groups with moderate obesity trained with moderate- or high- intensity interval training; GH: growth hormone; BL: blood lactate; HOMA-IR: homoeostasis model assessment index for insulin resistance.

**Table 4 children-10-01495-t004:** Percentage of change (Δ), in anthropometric and biochemical data (± standard deviation) between pre-test and post-test in each group.

	MOG_MI_ (*n* = 15) % Pre vs. Post	MOG_HI_ (*n* = 14) % Pre vs. Post	SOG_MI_ (*n* = 14) % Pre vs. Post	SOG_HI_ (*n* = 14) % Pre vs. Post
BM (kg)	−1.24 ± 0.8	−2.07 ± 0.8 ^§^	−1.78 ± 0.9	−1.94 ± 0.7 ^§^
BMI (kg·m^−2^)	−1.59 ± 0.5	−2.83 ± 1.1 ^£^	−2.46 ± 0.7	−2.19 ± 0.4
%BF (%)	−4.09 ± 0.9	−7.52 ± 0.8 ^£^	−2.73 ± 1.2	−4.18 ± 1.6
WC (cm)	−3.64 ± 1.1	−4.92 ± 1.2 ^£^	−2.64 ± 0.9	−3.56 ± 1.5
LBM (kg)	0.77 ± 0.1	2.36 ± 0.4 ^£^	0.79 ± 0.1	1.80 ± 0.3
GH (ng·mL^−1^)	2.94 ± 1.2	7.14 ± 1.5 ^£^	3.7 ± 1.4	3.57 ± 1.2
BL (mmol·L^−1^)	−1.66 ± 0.4	−7.5 ± 1.3 ^§^	−2.31 ± 0.5	−7.7 ± 1.2 ^§^
Glucose (mmol·L^−1^)	−5.61 ± 0.9	−7.95 ± 0.8 ^£^	−3.33 ± 0.5	−4.10 ± 0.9
INS (μU·mL^−1^)	−18.09 ± 1.7	−25.49 ± 1.9 ^§^	−18.39 ± 1.4	−25.34 ± 0.9 ^§^
HOMA-IR	−22.69 ± 1.1	−31.42 ± 0.9 ^§^	−21.12 ± 0.7	−28.88 ± 1.2 ^§^

Note: SOG_MI_ and SOG_HI_: the groups with severe obesity trained with moderate- or high-intensity interval training; MOG_MI_ and MOG_HI_: the groups with moderate obesity trained with moderate- or high-intensity interval training. BM: body mass; BMI: body mass index; %BF: percentage of body fat; WC: waist circumference; LBM: lean body mass, GH: growth hormone; BL: blood lactate; HOMA-IR: homoeostasis model assessment index for insulin resistance. Significantly different from the other groups: ^£^: *p* ˂ 0.05. Significantly different from the groups training at moderate intensity: (^§^): *p* ˂ 0.05.

## Data Availability

The data that support the findings of this study are available from the corresponding author, upon reasonable request.

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
