# Peer review of "High-Intensity Interval Training in Female Adolescents with Moderate or Severe Obesity"

_children, 2023, doi:10.3390/children10091495_

Round 1

Reviewer 1 Report

First of all, I would like to congratulate you on the work carried out, which addresses an interesting topic for the scientific community, committed to the relationship between children and sport. However, I allow myself to suggest improvements in the structure of the sections, with the aim of facilitating a more fluid understanding of the article.   In the summary section: They must add the number of total participants and not only by groups. On line 35 checking acronyms. Redrafting the results, they do not show that only HIIT improves the measured parameters. Add a concluding paragraph to conclude.   2.1. Participants: It is necessary to write the section or to improve the understanding of the choice of the final sample and the classification of the participants. On the one hand, the data of the total population could be exposed, and on the other, the data of the groups. Indicate the initial sample and the final sample after the exclusion of participants described in section 2.4. Add a specific section for inclusion and exclusion criteria in this section not a paragraph. They must also add a reference on line 100 2.2 Anthropometric measurements: They must indicate the measurement protocol followed for anthropometric measurements. I think the methodology used by you is the same as the one in this DOI article: https://doi.org/10.3390/app11115063. They must add a reference on line 124. 2.3 Blood measurements. They must add a reference on line 149 2.4 Group selection test: Paragraph 161 to 163 should be passed and worded within section 2.1 or as exclusion criteria.

Author Response

Answers to reviewers

            We thank the reviewers and the editor for their thorough review of our work and for the very constructive and helpful comments. We have taken the comments into consideration and have provided specific responses for each reviewer. Our responses in the manuscript appear in red typeface. We hope that this version has been improved and that is now suitable for publication in your journal. All authors have made sufficient contributions, responded to the comments, and have approved the submitted manuscript.

Furthermore, we are ready to make any further changes that would be deemed necessary for any deeper improvement.

Please find enclosed the revised version of the manuscript that we would like to submit to the Children journal.

Manuscript title:  High-Intensity Interval Training in Female Adolescents with Moderate or Severe Obesity

Dear Reviewer, 1,

We would like to thank you for the time allowed to this review process. Below, you can find our responses; each comment is followed by its respective reply. We made changes in the manuscript to address suggestions and make it clearer for the readers. Our responses in the manuscript appear in red typeface. We very much appreciate your comments on the document, which have helped us to improve its quality.

Best regards,

The Authors

Legend:

R1(Reviewer 1)

A (Authors)

First, I would like to congratulate you on the work carried out, which addresses an interesting topic for the scientific community, committed to the relationship between children and sport. However, I allow myself to suggest improvements in the structure of the sections, with the aim of facilitating a more fluid understanding of the article.   

R1 : In the summary section: They must add the number of total participants and not only by groups.

A : Thanks to the reviewer comment which helped to make clearer the groups repartition. This was added in red « Fifty-seven participants were divided into moderate obesity group (MOG, n=29) and severe obesity group (SOG, n=28). Two sub-groups were established to practice a HIIT and a MIIT programs (SOGHI, n= 14; SOGMI, n= 14), and (MOGHI, n= 14; MOGMI, n= 15). 

R1 : On line 35 checking acronyms.

A : We checked the acronym on line 35 which is (HRR). We should note that this was explained when it appeared first (heart rate reserve) as mentioned on line 33. We hope it is clearer now.

R1: Redrafting the results, they do not show that only HIIT improves the measured parameters.

A :As the reviewer commented, we agree that all groups increased their anthropometrical parameters. The higher increases were noted in the moderate obesity group training which trained at high intensity (MOGHI), this was further noted in the biological parameters which is clearer in Table 4 reporting the Percentage of change. To be more accurate we modified the conclusion part at the abstract section. Here is the change. “We may conclude that HIIT was the effective method to manage anthropometric and biological parameters; this was more pronounced in moderate obesity group confirming the priority to body-fat reduction.” We hope this modification makes it clearer.

R1 : Add a concluding paragraph to conclude.   

A : This was added as the reviewer suggested. Thanks, we think it is better now.

R1 : 2.1. Participants: It is necessary to write the section or to improve the understanding of the choice of the final sample and the classification of the participants. On the one hand, the data of the total population could be exposed, and on the other, the data of the groups. 

A : We agree that the presentation of the choice of groups was not clear enough neither their classification. We tried to reorganize this paragraph and add what is lacking to make it clearer to the reader. We hope that our modifications are in line to the reviewer's comment.

These are modifications: Fifty-seven obese girls (age 16.4 ± 0.8 years) were recruited. The participants were middle school students and were classified according to their body mass index (BMI) class from the algorithm provided by the Centers for Disease Control and Prevention. Each participant must have a BMI ≥ 97th percentile according to the French standards and a percentage of body fat (%BF) ≥ 30%. They were classified as moderately obese (MOG, n= 29) those who have a BMI between 30.0 and 34.9 kg·m-2, and severely obese (SOG, n= 28) those who have a BMI between 35.0 and 39.9 kg·m-2. In each BMI class, two sub-groups were established to practice a HIIT and a MIIT programs (SOGHI, n= 14; SOGMI, n= 14), and (MOGHI, n= 14; MOGMI, n= 15).

Means and standard deviation (SD) for BMI, %BF and body mass (BM) were: 34.1±1.6 kg·m-2, 38.6±3.9% and 95.1±4.5 kg, respectively.

R1 : Indicate the initial sample and the final sample after the exclusion of participants described in section 2.4. 

A: the initial sample and the final sample after the exclusion of participants was described more clearly as the reviewer suggested. This was added in section 2.4:  “Therefore, we mention that the initial sample included 62 participants (31 participants in each of SOG and MOG), three of them from SOG and 2 from MOG were excluded from the experimentation after having verified their ability to perform HIIT safely.”

R1 : Add a specific section for inclusion and exclusion criteria in this section not a paragraph. 

A : A specific section for inclusion and exclusion criteria was added to the manuscript as suggested. We hope it is better now.

R1 : They must also add a reference on line 100 

A : A reference was added as suggested. [18] Cole TJ, Bellizzi MC, Flegal KM, Dietz WH (2000) Establishing a standard definition for child overweight and obesity worldwide: international survey. BMJ 320:1240–1243.

R1 : 2.2 Anthropometric measurements: They must indicate the measurement protocol followed for anthropometric measurements.

A : The measurement protocol was added and the instruments used were cited. This was mentioned in section 2.2 : Body mass (BM) (to the nearest 0.1 kg), percentage of body fat (%BF) and lean body mass (LBM) were assessed by bioelectrical impedance analysis (TBF-300, Tanita®, Tokyo, Japan) and the body mass index was calculated (BMI = Mass [kg]/(Height [m])2 ). We hope it is better now.

R1 : I think the methodology used by you is the same as the one in this DOI article: https://doi.org/10.3390/app11115063.

A : As per the methodology followed in our study it was related to the main topic and some parts of it may have been used in other studies which we think is normal in the scientific research in order to reach the main goal. In fact, the study cited by the reviewer has a different subject and methodology which we think is interesting to read.

This is the study corresponding to the mentioned DOI: Impact of Kinanthropometric Differences According to Non-Professional Sports Activity Practiced

Daniel J. Navas Harrison, Ana María Pérez Pico and Raquel Mayordomo

R1 : They must add a reference on line 124. 

A : We respect the reviewer’s comment  concerning adding a reference which we think can help the reader, that’s why we tried to write directly the formula used and which will be also much useful to the reader. This was added:   the body mass index was calculated (BMI = Mass [kg]/(Height [m])2 ). We hope it is better now.

R1 : 2.3 Blood measurements. They must add a reference online 149 

A : A reference was added as suggested. « Guzzaloni G, Grugni G, Mazzilli G, Moro D, Morabito F. Comparison between B-cell function and insulin resistance indexes in pre-pubertal and pubertal obese children. Metabolism. 2002;51:1011–6. »

R1 : 2.4 Group selection test: Paragraph 161 to 163 should be passed and worded within section 2.1 or as exclusion criteria.

A : As requested the paragraph was moved to the inclusion exclusion criteria which we think is much better. Thanks to the reviewer for helping us to improve the quality of the manuscript.

Reviewer 2 Report

Dear Authors,
Thank you for the interesting research.

My main concerns are:
I would like to ask you to write very specifically whether the hypothesis raised at the beginning of the research was confirmed?

Did all the selected study girls participate in the study, or were there any who dropped out? It should be mentioned in the methodology section.

In section 2.1. I would recommend participants to describe in more detail how you invited the girls to the study, i.e. where did you get the girls BMI information from?  How did you divide the girls into groups?

You write: In each BMI class, two sub-groups were established to practice a HIIT and a MIIT programs, so I would like to check whether you have formed a control group?

Author Response

Answers to reviewers

         We thank the reviewers and the editor for their thorough review of our work and for the very constructive and helpful comments. We have taken the comments into consideration and have provided specific responses for each reviewer. Our responses in the manuscript appear in red typeface. We hope that this version has been improved and that is now suitable for publication in your journal. Furthermore, we are ready to make any further changes that would be deemed necessary for any deeper improvement.

Please find enclosed the revised version of the manuscript that we would like to submit to the Children journal.

Manuscript title:  High-Intensity Interval Training in Female Adolescents with Moderate or Severe Obesity

Dear Reviewer 2,

We would like to thank you for the time allowed to this review process. As a result, we are submitting the revised version for a possible publication in this respectable Journal. Below, you can find our responses; each comment is followed by its respective reply. We made changes in the manuscript to address suggestions and make it clearer for the readers. Our responses in the manuscript appear in red typeface. We very much appreciate your comments on the document, which have helped us to improve its quality.

All authors have made sufficient contributions, responded to your comments, and have approved the submitted manuscript.

Best regards,

The Authors

Legend:

R2(Reviewer 2)

A (Authors)

Reviewer :

Dear Authors,
R2 : Thank you for the interesting research.

A : We thank the reviewer for his positive opinion relating to our research, which gives us even more courage to broaden our scientific themes to add more in the field of research.

R2 : My main concerns are:
I would like to ask you to write very specifically whether the hypothesis raised at the beginning of the research was confirmed?

A : Thanks for this comment which helped us to add a paragraph at the end of the manuscript showing that our hypothesis was confirmed. This was added as the reviewer suggested. « At the end we emphasize that both exercise intensities (high and moderate) improved anthropometric parameters. However, the higher decrease in %BF was noted with HIIT as shown in (Table 4) in moderate obesity group, which means that decreasing body weight is essential in young obese, this may let them feel better after exercising. Thus, we presume that these feelings that appear after maximal exercise encourage more obese participants to adhere to training programs in the future, as has been reported [38]. Furthermore, our hypothesis raised at the beginning of this research may be confirmed with the important associations in the MOGHI shown between blood GH levels and %BF, WC, and blood insulin. ».

this reference was also added to the bibliography: Biddle SJH, Batterham AM. Highintensity interval exercise training for public health:a big HIT or shall we HIT it on the head? Int J Behav Nutr Phys Act. 2015;12:95. DOI: 10.1186/s12966-015-0254-9

R2 : Did all the selected study girls participate in the study, or were there any who dropped out? It should be mentioned in the methodology section.

A : Thanks to the reviewer, we think that this present an important point in the methodology section. However a paragraph was added in section 2.5 : “Therefore, we mention that the initial sample included 62 participants (31 participants in each of SOG and MOG), three of them from SOG and 2 from MOG were excluded from the experimentation after having verified their ability to perform HIIT safely”.

R2 : In section 2.1. I would recommend participants to describe in more detail how you invited the girls to the study, i.e., where did you get the girls BMI information from?  How did you divide the girls into groups?

A : This was added in the section 2.1 : « The participants were middle school students, were invited by their physical education teachers and were classified according to their body mass index. This BMI was determined by a school trained pediatrician. »

« To constitute the intervention groups, subjects were randomized and stratified (according to age and BMI). »

Thereafter participants were required to be able to perform a submaximal ramp incremental exercise test on a cycle ergometer otherwise they are not recruited for the study. This was cited in the inclusion exclusion criteria. We hope it is better now. Thanks to the reviewer constructive comments.

R2 : You write: In each BMI class, two sub-groups were established to practice a HIIT and a MIIT programs, so I would like to check whether you have formed a control group?

A : As the reviewer noted there is two sub-groups corresponding to the severe obesity group and to the moderate obesity group, which gave (severe obesity group training at HI and the other to  MI with no control group specified, but our analysis was cross-factored.

Thus to be more accurate we conducted a multi-group experimental study with independent variables (obesity levels and training intensity levels) to assess the effect of training on different levels of obesity. In this context, the two factors were "obesity levels" (with two levels: severe and moderate) and "training intensity levels" (with two levels: high and moderate). The aim was to analyze how these two factors influence the results of the study, for example, changes in weight, physical condition, and biological parameters.

We hope that we have responded to the author comment.

Reviewer 3 Report

I suggest that the authors use a better topic for the paper.

participants should be clarified more in detail. Inclusion and exclusion criteria, formular for sample size calculation, etc….

Please add some practical implications for the findings of this paper.

Author Response

Answers to reviewers

         We thank the reviewers and the editor for their thorough review of our work and for the very constructive and helpful comments. We have taken the comments into consideration and have provided specific responses for each reviewer. Our responses in the manuscript appear in red typeface. We hope that this version has been improved and that is now suitable for publication in your journal. Furthermore, we are ready to make any further changes that would be deemed necessary for any deeper improvement.

Please find enclosed the revised version of the manuscript that we would like to submit to the Children journal.

Manuscript title:  High-Intensity Interval Training in Female Adolescents with Moderate or Severe Obesity

Dear Reviewer, 3,

We would like to thank you for the time allowed to this review process. As a result, we are submitting the revised version for a possible publication in this respectable Journal. Below, you can find our responses; each comment is followed by its respective reply. We made changes in the manuscript to address suggestions and make it clearer for the readers. Our responses in the manuscript appear in red typeface. We very much appreciate your comments on the document, which have helped us to improve its quality.

All authors have made sufficient contributions, responded to your comments, and have approved the submitted manuscript.

Best regards,

The Authors

Legend:

R3 (Reviewer 3)

A (Authors)

Reviewer 3 :

R3 : I suggest that the authors use a better topic for the paper.

A : We thank the reviewer for this suggestion, We participated as invited for a Special Issues "10th Anniversary of Children: Urgent Issues in the Health of Children – 2023–2033" after a preliminary evaluation by Editor in Chief.

R3 : participants should be clarified more in detail. Inclusion and exclusion criteria, formular for sample size calculation, etc.….

A : As suggested by the reviewer, we gave more clarifications related to the participants and we gave more details about the inclusion and exclusion criteria.

Concerning the sample size: Generally, the sample size for any study depends on the acceptable level of significance, power of the study, effect size and the standard deviation in the population of the study. There are several methods used to calculate the sample size depending on the type of data or study design. In our study, the sample size is calculated using the following formula:

Z is a constant (set by convention according to the accepted α error). In this study, we are in 5% α error with 80% power (the most of studies accept a power of 80%. This means that we are accepting 20% we will miss a real difference). Zα is 1.96 and Z1−β is 0.8416. The estimated statistical variance σ2 (based on the previous studies the main variables are physiological and biological variables). For Δ, the effect size would be 0.15.       

References:

-Kirby A, Gebski V, Keech AC. Determining the sample size in a clinical trial. Med J Aust 2002;177:256-7.

- Larsen S, Osnes M, Eidsaunet W, Sandvik L. Factors influencing the sample size, exemplified by studies on gastroduodenal tolerability of drugs. Scand J Gastroenterol 1985;20:395-400

R3 : Please add some practical implications for the findings of this paper.

A : Some practical implications were added as the reviewer suggested. We think that this has improved the manuscript. This was added:

« At the end we emphasize that both exercise intensities………………........Furthermore, our hypothesis raised at the beginning of this research may be confirmed with the important associations in the MOGHI shown between blood GH levels and %BF, WC, and blood insulin. »

Reviewer 4 Report

Dear authors

Please review the document.

Thank you.
